# Rethinking Parity Check Enhanced Symmetry-Preserving Ansatz

**Ge Yan, Mengfei Ran, Ruocheng Wang, Kaiseng Pan, Junchi Yan**[*]
Dept. of CSE & School of AI & Zhiyuan College, Shanghai Jiao Tong University
{yange98,rmf2021,Ignite,pks0813,yanjunchi}@sjtu.edu.cn

## Abstract

With the arrival of the Noisy Intermediate-Scale Quantum (NISQ) era, Variational Quantum Algorithms (VQAs) have emerged to obtain possible quantum advantage. In particular, how to effectively incorporate hard constraints in VQAs remains a critical and open question. In this paper, we manage to seamlessly combine the Hamming Weight Preserving ansatz with a topological-aware parity check on physical qubits to enforce error mitigation and further hard constraints. We demonstrate such a combination significantly outperforms peer VQA methods on both quantum chemistry problems and constrained combinatorial optimization problems e.g. Quadratic Assignment Problem. Our extensive experimental results on both simulators and superconducting quantum processors verify that the combination of HWP ansatz with parity check is among the most promising candidates to show quantum advantages in the NISQ era to solve more realistic problems.

## 1 Introduction

Variational Quantum Algorithms (VQAs) [17, 73] and their derivatives [80] have garnered increasing attention as numerous studies investigate their potential to achieve quantum supremacy. With the advent of the NISQ era [65, 10] and the improved deployment capability [77], the exploration of new VQAs has accelerated, as these algorithms have shown promise in delivering quantum advantage on near-term quantum devices [17]. However, despite their potential, commonly used VQAs such as the QAOA [27] for Quadratic Unconstrained Binary Optimization (QUBO) and UCCSD [67] for ground state energy estimation are not inherently designed to handle hard constraints. Typically, these constraints are modeled as soft penalty terms within the objective function, which may not be the most efficient approach. Addressing the natural incorporation of symmetries and hard constraints directly into VQAs remains an open and critical challenge for advancing the field.

In this paper, we investigate Hamming Weight preserving (HWP) ansatz [78] and parity check [53] as a novel approach for error mitigation and the imposition of hard constraints in quantum circuits, rather than modeling these constraints as penalty terms in the Hamiltonian, as is common in the literature [18, 48]. The HWP ansatz, operating in a constrained subspace, utilizes parameterized gates that maintain the number of non-zero elements in the quantum state. Recent work [78] has thoroughly analyzed the expressivity and trainability of the HWP ansatz, demonstrating its outstanding performance in capturing fundamental symmetries, such as total spin conservation. Meanwhile, quantum LDPC code [53, 13] and surface code [29] have been widely widely adopted in quantum error correction through the use of stabilizers to detect and correct qubit errors. However, a significant challenge remains in identifying quantum ansatze that naturally facilitate error correction mechanisms. In response, we propose a combined framework that integrates the HWP ansatz with parity check operations. Since the HWP ansatz inherently preserves the number of non-zero elements in quantum states, it offers a promising foundation for enhancing parity check performance in mitigating errors. By employing parity checks as projective measurements, quantum states can be

---

[*]Correspondence author. This work was partly supported by NSFC (92370201, 62222607) and Shanghai Municipal Science and Technology Major Project under Grant 2021SHZDZX0102.

constrained directly to the problem subspace, thereby introducing a robust mechanism for error mitigation while expanding the utility of parity checks beyond their conventional role. This approach allows for more effective error correction and constraint enforcement within quantum circuits.

In this paper, we first analyze the effect of utilizing parity check as an error mitigation method for HWP ansatz on popular quantum chemistry problems, whose (symmetry) constraints can be effectively addressed by HWP. The parity check block is constructed by a cascade of CNOT gates which only requires nearest neighbor connectivity of physical qubits. By repeatedly inserting parity check blocks in the quantum circuit we can mitigate the influence of unexpected bit-flip errors. The results on noiseless simulator demonstrate that the proposed universal HWP ansatz is able to exceed UCC ansatz with single, double, and even triple excitation. We also test the performance of parity checks with HWP ansatz against other symmetry verification (SV) methods [38, 70] with UCC ansatze. Under the simulated noise, we observe that only verifying the number of non-zero elements at the end of the circuit is not enough to determine whether there is an occurrence of error, so constantly parity checks in the circuit are essential for detecting bit-flip errors.

We then combine HWP ansatz and parity check to develop an efficient paradigm to incorporate additional hard constraints beyond the capability of HWP, to enable solving other constrained problems in classic computing, specifically Quadratic Assignment Problem (QAP) [55]. This problem is known an NP-hard combinatorial optimization (CO) problem, as widely studied in literature in both classic machine learning (ML) [79] and quantum ML [80]. It aims to find an optimal permutation matrix with each column and row having only one non-zero element. Specifically, we map the permutation matrix to the physical qubit lattice topology with each qubit connected to its four nearest neighbors. We then apply HWP layers on the rows to ensure a smaller subspace with parity checks on the columns appearing at intervals to further restrict the states to QAP subspace. The final loss is calculated with in-constraint states so that we can find the optimal solution within the constraints.

To further illustrate the capability and efficiency of the proposed approach, we conduct experiments on both simulators and superconducting quantum processors. We compare a wide range of baseline methods with soft constraints, e.g., HEA [46], QAOA [27], XYmixer [39], and we also add hard constraints to some of them with the proposed paradigm. The numerical results on the simulator demonstrate the outstanding performance of the proposed hard constraint paradigm. We also test our methods on the Traveling Salesman Problem (TSP) to illustrate the capability of our methods on other CO problems by reducing it into QAP. For the hardware experiments on a superconducting quantum processor, the proposed method also show promising performance, especially with the physical qubit topology (no SWAP gates required in the compilation). We move the related works and preliminaries to the Appendix. **The contributions of this paper are:**

1) We illustrate how to utilize parity check to mitigate quantum errors and incorporate further constraints for HWP ansatz. The combination of these two is among the most promising candidates to demonstrate quantum advantages in the NISQ era (for VQA) to solve more realistic problems.

2) We discuss the connection between HWP and UCC ansatze on quantum chemistry problems and further examine the efficiency of parity check as error mitigation on the simulator with noise. Results illustrate that universal two-qubit HWP gates exceed UCC with high-order excitation terms and parity check in between HWP gates can mitigate errors better than SV [38, 70].

3) We propose a novel hard-constraint VQA with parity check as further constraints for HWP. We map the permutation matrix in QAP to the physical qubit topology and enable the maximum utility of qubit connectivity. The superior performance over peer VQAs on both simulator and quantum processor shows the capability and efficiency of our method on constrained CO, e.g. QAP and TSP.

## 2 Parity Check as Error Mitigation for HWP

We first illustrate how to use parity check as error mitigation for HWP ansatz. The HWP ansatz maintains the number of non-zero elements in the whole unitary transformation, which makes it easy to detect any unexpected bit-flip. By constantly applying parity checks we can correct these errors.

### 2.1 Ground State Energy Estimation

The ground state energy estimation problem, which is the very first step in computing the energetic properties of molecules and materials, has received intensive attention with various VQE approaches [73]. The ground state of a molecule is the stationary state with the lowest allowed energy, which can be estimated given the types and relative coordinates of the atoms in the molecule and the

number of orbitals and electrons. Providing a molecular Hamiltonian $\mathcal{H}_m$, and a trial wave function $|\psi\rangle$, the ground state energy $E_0$ is bounded by [28]:

$$E_0 \leq \frac{\langle\psi|\,\mathcal{H}_m\,|\psi\rangle}{\langle\psi|\psi\rangle}, \tag{1}$$

where the equality holds if and only if the parameterized wavefunction $|\psi\rangle$ is the ground state. To solve this problem on a quantum computer, we need to design the ansatz wavefunction, which is bound to be unitary operations since we are operating on a quantum computer and all the quantum gates are unitary transformations. We then describe the unitary parameterized ansatz as $\mathbf{U}(\boldsymbol{\theta})$. The qubits are initialized as $|0\rangle^n$ (abbreviated as $|\mathbf{0}\rangle$) with $n$ as the number of orbitals under the Jordan-Wigner transformation [45]. Notice that any quantum state is necessarily a normalized wavefunction, so the cost function of the VQE problem is [73]:

$$E_{VQE} = \min_{\boldsymbol{\theta}} \langle\mathbf{0}|\,\mathbf{U}^\dagger(\boldsymbol{\theta})\mathcal{H}_m\mathbf{U}(\boldsymbol{\theta})\,|\mathbf{0}\rangle. \tag{2}$$

The molecular Hamiltonians often come with symmetry constraints, and we can utilize HWP ansatz to reduce the evolving space and draw support from parity check as an error mitigation method.

## 2.2 HWP ansatz for Quantum Chemistry

We first introduce two basic models, namely the Fermi-Hubbard model [44] and Unitary Coupled Cluster model (UCC) [64], to show how HWP ansatz can be linked to quantum chemistry. Both models describe the hopping of electrons on orbitals (UCC model) or sites (Hubbard model) by creation and annihilation operators (see definition in Apx. B.2). We have the following observation:

*Remark* 2.1. Both hopping terms in the UCC and Fermi-Hubbard model are HWP operators.

Recall the hopping term on adjacent sites $i$ and $j$ in the Fermi-Hubbard model is defined as:

$$\mathcal{H}_{FH} = a_i^\dagger a_j + a_j^\dagger a_i = \begin{pmatrix} 0 & 0 & 0 & 0 \\ 0 & 0 & 1 & 0 \\ 0 & 1 & 0 & 0 \\ 0 & 0 & 0 & 0 \end{pmatrix} = \sigma_x \otimes \sigma_x + \sigma_y \otimes \sigma_y. \tag{3}$$

where $a$ and $a^\dagger$ are the annihilation and creation operators, respectively (see detailed definition in Apx. B.2). Similarly, the cluster operator in the coupled cluster theory is $T = a_i^\dagger a_j$. For the state transformation for UCC model, we follow the form $|\psi\rangle = e^{T-T^\dagger}|\psi_0\rangle$ [5], where $T - T^\dagger$ is an anti-Hermitian operator which makes it suitable for quantum computers since the exponential of an anti-Hermitian operator is a unitary operator. The Hamiltonian for the single excitation term is

$$\mathcal{H}_{UCC} = \frac{1}{i}\left(T - T^\dagger\right) = \frac{1}{i}\begin{pmatrix} 0 & 0 & 0 & 0 \\ 0 & 0 & 1 & 0 \\ 0 & -1 & 0 & 0 \\ 0 & 0 & 0 & 0 \end{pmatrix} = \begin{pmatrix} 0 & 0 & 0 & 0 \\ 0 & 0 & -i & 0 \\ 0 & i & 0 & 0 \\ 0 & 0 & 0 & 0 \end{pmatrix}. \tag{4}$$

Both of the above Hamiltonian fits the definition of HWP (see Eq. 21 in Apx.B for the introduction of the preliminaries of HWP), so they are both HWP operators.

**Lemma 2.2.** *The hopping terms in UCC and Fermi-Hubbard are not universal in the HWP subspace.*

According to Theorem B.1 [66] (see details in Apx. B), an HWP operator with a given connectivity is universal if and only if the dimension of the corresponding dynamical lie algebra (DLA) is $d_k^2$ with $d_k = \binom{n}{k}$, where $n$ and $k$ is the number of orbitals and electrons, respectively. For nearest neighbor (NN) connectivity, we can derive the dimensions of DLA for operators in Eq. 3 and Eq. 4 as follows:

$$dim_{nn}(\mathfrak{g}_{UCC}) = \frac{1}{2}n(n-1), \quad dim_{nn}(\mathfrak{g}_{FH}) = \begin{cases} (n+1)(n-1) & n \text{ is odd} \\ \frac{1}{2}n(n-1) & n \text{ is even} \end{cases} \tag{5}$$

For fully connected (FC) connectivity, the dimensions of DLA for the two operators are:

$$dim_{fc}(\mathfrak{g}_{UCC}) = \frac{1}{2}d_k(d_k-1), \quad dim_{fc}(\mathfrak{g}_{FH}) = \begin{cases} (d_k+1)(d_k-1) & n \neq 2k \\ \frac{1}{2}(d_k+2)(d_k-2) & n = 2k \end{cases} \tag{6}$$

All of the dimensions do not meet the requirement of $d_k^2$, so both Eq. 5 and Eq. 6 are not universal in the HWP subspace, which aligns with the fact that Eq. 3 and Eq. 4 are truncated hopping terms.

**Theorem 2.3.** *An ansatz $\mathbf{U}(\boldsymbol{\theta})$ can solve the ground state energy estimation problem without truncation if the ansatz is universal under the $d_k$-dimensional HWP subspace.*

The detailed proof is provided in Apx. C. In contrast to the Fermi-Hubbard model that does not have higher-order hopping terms, the UCC model has double and triple excitation operators [41] able to improve the accuracy of the final ground states. However, the double and triple excitation operators require 4-qubit and 6-qubit gates respectively, which makes it unaffordable when decomposing them into basic gates. Thus, we seek two-qubit HWP gates (much easier to implement than double and triple excitation operators) that are universal under the HWP subspace with no truncation at all.

**Definition 2.4.** We propose an HWP gate namely NBS with the Hamiltonian and dimension of DLA:

$$\mathbf{H}_{NBS} = \begin{pmatrix} 0 & 0 & 0 & 0 \\ 0 & 1 & \mathrm{i} & 0 \\ 0 & -\mathrm{i} & 1 & 0 \\ 0 & 0 & 0 & 0 \end{pmatrix}, \quad \dim(\mathfrak{g}) = \begin{cases} d_k^2 & n \neq 2k, \\ d_k^2/2 - 1 & n = 2k. \end{cases} \tag{7}$$

NBS gate is very close to universal under NN connectivity and simpler than the one proposed in [78]. Thus, we can use the NBS gate to construct an NN connected ansatz without any truncation (UCCSD [67]) or prior knowledge about quantum chemistry (adaptVQE [36]).

A common error mitigation technique in VQE for the UCC and Hubbard model is SV [70, 38], which discards runs where the final and initial occupations do not match. This method leverages the fact that these VQE ansatze exhibit symmetries such as number conservation per spin sector and time-reversal symmetry. Specifically, when verifying the symmetry of total spins, one counts the number of non-zero elements in the final state to detect any unexpected bit-flip or readout errors that may alter the total spin count. However, SV only reflects symmetries in the final state and requires readout of all qubits (or at least an equal number of measurements), adding complexity.

To simplify both the verification of final states and the intermediate states during computation, we propose utilizing parity checks, a technique commonly employed in classical and quantum error correction. Unlike SV, which counts the number of non-zero elements in the quantum state, parity checks only evaluate the parity of these non-zero elements. While a single parity check extracts less information than SV, applying parity checks continuously throughout the quantum circuit ensures that the final state retains the same Hamming Weight as the initial state (as long as the probability of multiple bit-flips occurring between two parity checks is sufficiently low). The

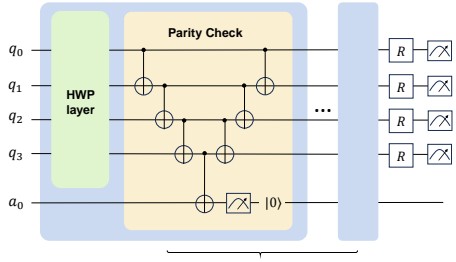

Figure 1: The overall circuit structure for parity checks and HWP ansatz.

HWP ansatz is an ideal complement to parity checks because it guarantees a constant HW for the intermediate states, allowing for the seamless integration of parity checks at any point in the ansatz. This approach not only mitigates errors but also simplifies the error detection process by embedding it directly into the circuit, as illustrated in Fig. 1.

### 2.3 Simulated Experiments on State Preparation

**Dataset:** We select three well-studied molecules, i.e. Hydrogen ($H_2$), Lithium Hydride (LiH), and Water ($H_2O$). The molecular Hamiltonian is obtained from the Python package OpenFermion [58]. The computational basis for all the molecules is STO-3G with Jordan-Wigner transformation. To simulate the circuit with noise on Qiskit, we utilize the Aer-simulator from Qiskit based on the density

Table 1: Statistics of molecules. $n$, $k$, $d_k$ is the number of orbitals, electrons, and HWP subspace dimension respectively.

| Molecules | $H_2$ | LiH | $H_2O$ |
|-----------|-------|-----|--------|
| $n$ | 4 | 8 | 8 |
| $k$ | 2 | 2 | 4 |
| $d_k$ | 6 | 28 | 70 |

matrix which is time-consuming. Therefore, we freeze some of the inactive orbitals to reduce the problem size of the above molecules. The detailed molecular information is listed in Tab. 1

**Baselines:** To show the efficiency of the HWP ansatz and the superiority of combining HWP ansatz with parity check, we select the well-studied UCC ansatz as our baselines. To better illustrate that universal HWP ansatz is able to solve the state preparation problem without any truncation, we include single excitation (UCCS), double excitation (UCCSD), and triple excitation (UCCSDT). All

Table 2: Numerical results for state preparation. "Error" stands for the energy error with respect to FCI energy, "prob" represents the success rate in SV and PC, "SE" stands for value less than $10^{-10}$.

| Method | Setting | $H_2$ Energy (Ha) | Error | Prob | LiH Energy (Ha) | Error | Prob | $H_2O$ Energy (Ha) | Error | Prob |
|---|---|---|---|---|---|---|---|---|---|---|
| UCCS | noiseless | -1.1173488878±2.30×10⁻⁸ | 1.88×10⁻² | — | -7.8618641468±1.29×10⁻⁷ | 1.82×10⁻³ | — | -74.3608375081±2.04×10⁻¹ | 3.76×10⁻¹ | — |
| | noise | -0.9176263520 | 2.19×10⁻¹ | — | -7.6984769446 | 1.65×10⁻¹ | — | -74.1416601629 | 5.95×10⁻¹ | — |
| | SV | -0.9706892814 | 1.66×10⁻¹ | 0.865 | -7.7302653047 | 1.33×10⁻¹ | 0.701 | -74.1720832569 | 5.65×10⁻¹ | 0.649 |
| UCCSD | noiseless | -1.1361893704±1.06×10⁻⁷ | 3.91×10⁻¹⁰ | — | -7.8629191864±3.61×10⁻⁴ | 7.62×10⁻⁴ | — | -74.7359393767±1.97×10⁻² | 9.85×10⁻⁴ | — |
| | noise | -0.5626316869 | 5.74×10⁻¹ | — | -6.8652894300 | 9.98×10⁻¹ | — | -73.8588494832 | 8.78×10⁻¹ | — |
| | SV | -0.7042719212 | 4.32×10⁻¹ | 0.716 | -6.8376290127 | 1.03×10⁰ | 0.500 | -73.8500427310 | 8.87×10⁻¹ | 0.501 |
| UCCSDT | noiseless | — | — | — | — | — | — | -74.7368968448±1.87×10⁻² | 2.79×10⁻⁵ | — |
| | noise | — | — | — | — | — | — | -73.8454283537 | 8.91×10⁻¹ | — |
| | SV | — | — | — | — | — | — | -73.8632794220 | 8.79×10⁻³ | 0.500 |
| Ours | noiseless | -1.1361894537±3.82×10⁻¹⁶ | SE | — | -7.8636816249±1.44×10⁻¹² | SE | — | -74.7369247415±1.83×10⁻¹⁰ | 1.59×10⁻¹⁰ | — |
| | noise | -0.6523354546 | 4.84×10⁻¹ | — | -7.1394207947 | 7.24×10⁻¹ | — | -73.9364705694 | 8.00×10⁻¹ | — |
| | SV | -0.7599740203 | 3.76×10⁻¹ | 0.747 | -7.1312360044 | 7.32×10⁻¹ | 0.505 | -73.9089583257 | 8.28×10⁻¹ | 0.505 |
| | SV+PC | -0.7697837117 | 3.66×10⁻¹ | 0.674 | -7.2627704120 | 6.01×10⁻¹ | 0.073 | -74.0486388806 | 6.88×10⁻¹ | 0.072 |

the ansatze are implemented with Qiskit-Nature [24] and initialized with Hartree-Fock state. The optimizer is SLSQP.

**Results on Simulators** The sensitivity analysis on the number of parity checks and hyperparameter settings are listed in Apx. E. We provide the results on the simulator with noise in Tab. 2. We first focus on the results of UCC ansatze with different excitation and the proposed HWP ansatz without noise. Notice that $H_2$ and LiH only have 2 active electrons so it is impossible to apply triple excitation in the ansatz. It is shown that adding high-order excitation terms in the UCC ansatz can improve the results, and the proposed universal HWP ansatz is able to solve the problem with no truncation at all which leads to energy with error less than $1 \times 10^{-10}$. The circuit depth of HWP ansatz is also much less than UCCSD and UCCSDT. Detailed comparisons of the circuit statistics are listed in Tab. 8

We make several important observations on the results with noise and error mitigation methods. 1) UCCS outperforms other methods with noise since it is extremely shallow. 2) SV with deep circuit is useless since the parity of the output state is approximate to 0.5, indicating the results for each qubit are close to a uniform superposition state. 3) More shots per Pauli string would be beneficial to better illustrate the performance of error mitigation methods. 4) Parity check can improve the results on deep ansatz. By combining other error mitigation approaches (such as readout error mitigation), we may further improve the accuracy. Therefore, we can conclude that parity check is an efficient error mitigation method for HWP ansatz, which can improve the result quality.

## 3 Parity Check as Further Constraints for QAP

Apart from the quantum chemistry problems, the HWP ansatz is proved to be able to serve as a hard constraint for combinatorial optimization problems [78]. In this section, we will demonstrate how to utilize parity check to enforce additional hard constraints for HWP ansatz so that we are capable of solving more complicated constraints.

### 3.1 Quadratic Assignment Problem

The Quadratic Assignment Problem (QAP) is a well-studied NP-hard problem dated back to [52]. A typical QAP instance of size $m$ is given by two matrices $\mathbf{F} \in \mathbb{R}^{m \times m}$, $\mathbf{D} \in \mathbb{R}^{m \times m}$, defining the flows between facilities and the distances between locations [55]. Its objective with constraints is:

$$\min \sum_{i,j=1}^{m} \sum_{k,p=1}^{m} \mathbf{F}_{ij} \mathbf{D}_{kp} \mathbf{X}_{ik} \mathbf{X}_{jp} \qquad \text{s.t.} \sum_{i=1}^{m} \mathbf{X}_{ij} = 1, \sum_{j=1}^{m} \mathbf{X}_{ij} = 1, \ 1 \leqslant i,j \leqslant m, \qquad (8)$$

where $\mathbf{X} \in \{0,1\}^{m \times m}$ is the permutation matrix illustrated in Fig. 2(a). The essence of QAP is to find the best permutation matrix which minimizes the objective function. Further, we define $\mathbf{W} \in \mathbb{R}^{m^2 \times m^2}$ as the energy matrix with $\mathbf{W} = \mathbf{F} \otimes \mathbf{D}$, corresponding to the vector product form of the flow and distance matrices. The QUBO form of QAP is:

$$\min \ \text{vec}(\mathbf{X})^{\top} \mathbf{W} \ \text{vec}(\mathbf{X}) \qquad \text{s.t.} \sum_{i=1}^{m} \mathbf{X}_{ij} = 1, \sum_{j=1}^{m} \mathbf{X}_{ij} = 1, \ 1 \leqslant i,j \leqslant m, \qquad (9)$$

where $\text{vec}(\mathbf{X}) \in \{0,1\}^{m^2 \times 1}$ denotes a vector by concatenating the columns of matrix $\mathbf{X}$ [79]. We can further derive the Hamiltonian of the Ising form of QAP:

$$\mathbf{H}_{QAP} = \frac{1}{4} \sum_{i,j=1}^{m} \sum_{k,p=1}^{m} \mathbf{D}_{ij} \mathbf{F}_{kp} (I - \sigma_{ik}^{z})(I - \sigma_{jl}^{z}), \qquad (10)$$

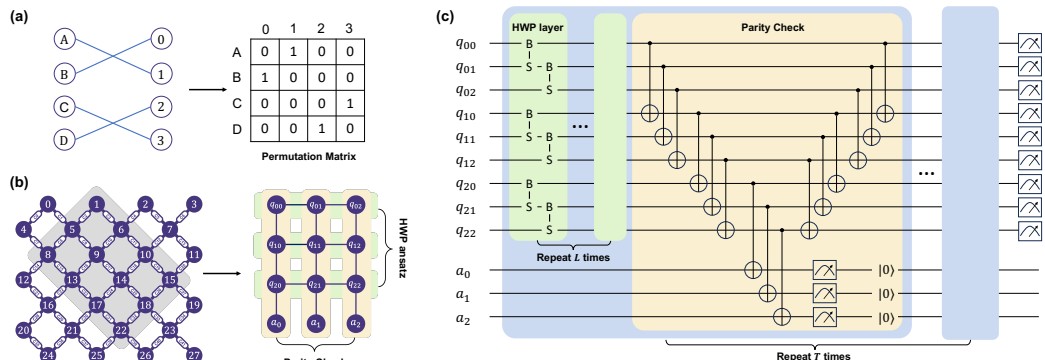

Figure 2: (a) The QAP instance with four facilities $\{A, B, C, D\}$ mapped to four locations $\{0, 1, 2, 3\}$. The permutation matrix encodes the mapping and satisfies the constraint that each facility is mapped to only one location and vice versa. (b) The topology of the physical qubits of superconducting quantum processor. For QAP with $m = 3$, we select $n = m^2$ working qubits denoted as $q_{ij}$ and $m$ ancilla qubits denoted as $a_i$ with the structure shown on the right. Each qubit $q_{ij}$ maps to the element $\mathbf{X}_{ij}$ in the permutation matrix and suffice the constraints that each row and column has only one $|1\rangle$. (c) The overall circuit for QAP with all the working and ancilla qubits is named the same as in (b).

where $\mathbf{H}_{QAP} \in \mathbb{R}^{2^n \times 2^n}$, and $\sigma^z$ is the pauli-z matrix. Thus, we need $n = m^2$ qubits to solve QAP.

### 3.2 HWP ansatz for Combinatorial Optimization

The QAP can be seen as optimizing a matrix $\mathbf{X} \in \{0, 1\}^{m \times m}$ with each row and each column having only one non-zero element. We can easily map the permutation matrix to the topology of the superconducting qubits as illustrated in Fig. 2(b). The coupler denoted as $c_i$ between two qubits stands for the allowance of two-qubit gates between the corresponding two qubits. It shows a commonly used topology [3, 34] for superconducting quantum processors with each qubit connected to its four nearest neighbors through couplers. This kind of topology on the NISQ device can produce better connectivity than the nearest neighbor connectivity for logical qubits where each logical qubit is connected to only two nearest neighbors. Therefore, we aim to make full use of the qubit topology to provide an algorithm that is suitable for those existing superconducting quantum processors.

Notice that if we map each element in matrix $\mathbf{X}$ to a qubit, we can easily adopt the HWP ansatz to meet the constraints of QAP. Each row and column of the qubits have exactly one $|1\rangle$ with the rest as $|0\rangle$, which can be converted to an HWP problem with dimension as $d_k = \binom{m}{1}$. Without loss of generality, we apply $m$ independent HWP ansatz on the row and we utilize a projective measurement on the column to ensure the post-measurement states are feasible states for QAP. Here we will introduce how to use parity check circuit to implement the projective measurement with only one ancilla qubit for each column. Since each column has exactly one $|1\rangle$, a sufficient and necessary condition for QAP is that the parity check results for all $m$ columns are all odd.

A detailed circuit for parity check is illustrated in Fig. 2(c). HWP ansatz is applied on each row of physical qubits and parity check is applied on each column. We repeat the NN connected HWP layers for $L$ times and then apply a parity check on each ancilla qubit. After measuring the ancilla qubits, we flip the working qubits back and reset the ancilla qubits as $|0\rangle$. The whole block is repeated $T$ times before we measure all the working qubits and calculate the loss. For the working qubits, the initial state should be a trivial state in the QAP subspace that can be easily prepared such as the identity permutation with qubit $q_{ii}$ as $|1\rangle$ and the rest as $|0\rangle$. The overall parameterized evolution can be written in the following unitary transformation:

$$\mathbf{U}_{QAP}(\boldsymbol{\theta}) = \prod_{t=1}^{T} \left( \mathcal{P} \times \prod_{l=1}^{L} \mathbf{U}_{HWP}(\boldsymbol{\theta}_{t,l}) \right), \tag{11}$$

where $\mathbf{U}_{HWP}$ is the unitary of the HWP layer with $\boldsymbol{\theta}_{t,l}$ as the parameters in block $t$ layer $l$, and $\mathcal{P}$ denotes the projective measurement by parity check. We utilize a cascade of CNOT gates to transfer the parity information from working qubits to the ancilla qubits, which will not further include SWAP gates in execution. Note that the parity check measurement on the ancilla qubits will not destroy

Table 3: Results on QAP simulation with the best in bold and the best in soft constraints underlined.

| CONSTRAINT | METHOD | $dim$ | $\eta$ | | | | $p_{optimal}$ | | | |
|---|---|---|---|---|---|---|---|---|---|---|
| | | | $m=3$ | $m=4$ | $m=5$ | $m=6$ | $m=3$ | $m=4$ | $m=5$ | $m=6$ |
| SOFT | HEA [46] | $2^n$ | 0.7000 | 0.0000 | — | — | 0.1746 | 0.0000 | — | — |
| | QAOA [27] | $2^n$ | 0.4715 | 0.5491 | — | — | 0.2497 | 0.0000 | — | — |
| | XYMIXER-NN [39] | $\binom{n}{m}$ | 0.5672 | 0.6927 | — | — | 0.1998 | 0.0999 | — | — |
| | XYMIXER-FC [39] | $\binom{n}{m}$ | 0.9957 | 0.4707 | — | — | 0.8738 | 0.1196 | — | — |
| | NBS-NN | $\binom{n}{m}$ | 0.7969 | 0.6046 | — | — | 0.5495 | 0.0500 | — | — |
| | NBS-FC | $\binom{n}{m}$ | 0.9975 | 0.4517 | — | — | 0.8765 | 0.0788 | — | — |
| HARD | XYMIXER-HARD [39] | $m!$ | **1.0000** | 0.9931 | 0.9837 | 0.9474 | **1.0000** | 0.8498 | 0.7500 | 0.3500 |
| | NBS-HARD | $m!$ | **1.0000** | **0.9991** | **0.9919** | **0.9826** | **1.0000** | **0.9448** | **0.8500** | **0.5000** |

the in-constraint quantum states on the working qubits, so we are able to restrict the states without collapsing the whole quantum system. After measuring the ancilla qubits, we flip back all the working qubits and reset the ancilla qubits as $|0\rangle$.

The parity checks can provide certain entanglement on the topological columns of qubits. The quantum states on the working qubits before the first parity check is $m$ independent pure states denoted as $|\psi_i\rangle$ with $i \in [0, m)$ on each row. The quantum state can be written as:

$$\boldsymbol{\delta}_1 = |0\rangle^{\otimes m} \left( \bigotimes_{i=0}^{m-1} |\psi_i\rangle \right) \left( \bigotimes_{i=0}^{m-1} |\psi_i\rangle \right)^{\dagger} \langle 0|^{\otimes m}, \tag{12}$$

where $\boldsymbol{\delta}_1$ is a density matrix for block 1 before parity check. After the first parity check, the states on the working qubits are transformed from $m$ independent quantum states on the rows to feasible states in the QAP subspace and other entangled states outside the subspace.

$$\boldsymbol{\rho}_1 = \sum_{i=0}^{2^m-1} |i\rangle \boldsymbol{\rho}_1(i) \langle i|, \tag{13}$$

where $\boldsymbol{\rho}_1$ is the density matrix for block 1 after parity check, and $|i\rangle$ denotes the quantum states on the ancilla qubits. We can see that the measurement changes the basis of $\boldsymbol{\delta}_1$ and $\boldsymbol{\rho}_1$, resulting feasible states for QAP in $\boldsymbol{\rho}_1$. Similarly, we denote the quantum state after the final parity check as

$$\boldsymbol{\rho}_T = \sum_{i=0}^{2^m-1} |i\rangle \boldsymbol{\rho}_T(i) \langle i|. \tag{14}$$

Since the quantum states on the working qubits are in QAP subspace if and only if the ancilla qubits are all measured as $|1\rangle$, the feasible final state on the working qubits is $\boldsymbol{\rho}_T(2^m - 1)$. The loss is:

$$\mathcal{L}_{QAP} = \mathrm{Tr}\left[ \mathbf{H}_{QAP} \times \boldsymbol{\rho}_T(2^m - 1) \right], \tag{15}$$

where $\mathbf{H}_{QAP}$ is the Hamiltonian of QAP with definition in Eq. 10. We only update the parameters based on those in-constraint states so that we are able to further find the optimal answer in the QAP subspace. The expectation value of obtaining $\boldsymbol{\rho}_T(2^m - 1)$ is

$$\mathrm{E}(p_{\boldsymbol{\rho}_T(2^m-1)}) = \frac{m!}{m^m}, \tag{16}$$

where $p_{\boldsymbol{\rho}_T(2^m-1)}$ denotes the probability of obtaining $\boldsymbol{\rho}_T(2^m - 1)$ from all the possible states. We can further derive the equation using the Stirling's formula and we have

$$\mathrm{E}(p_{\boldsymbol{\rho}_T(2^m-1)}) \approx \frac{\sqrt{2\pi m}(\frac{m}{e})^m}{m^m} = \frac{\sqrt{2\pi m}}{e^m}. \tag{17}$$

The expectation of obtaining feasible states decreases exponentially with $m$. However, this order of the expectation value is acceptable as the Ising model for QAP requires $n = m^2$ qubits. We will further show in the experiments that we generally have a better probability of obtaining feasible states and this is much better than using soft constraints.

### 3.3 Simulated Experiments on Quadratic Assignment Problem

**Dataset:** We generated a dataset comprising random instances of QAPs, with 100 instances for each size $m = \{3, 4, 5, 6\}$. Each instance includes a $m \times m$ distance matrix $\mathbf{D}$ and a flow matrix $\mathbf{F}$,

with elements $\mathbf{D}_{ij} = \mathbf{D}_{ji}$ and $\mathbf{F}_{ij} = \mathbf{F}_{ji}$, drawn from a uniform distribution $[0, 1]$. $\mathbf{D}_{ij} = \mathbf{D}_{ji} \sim U(0,1)$, $\mathbf{F}_{ij} = \mathbf{F}_{ji} \sim U(0,1), i \neq j$.

**Baselines.** *HEA [46]:* the most commonly used quantum machine learning model with a simple structure and only nearest neighbor connectivity is required; *QAOA [27]:* the originator of utilizing VQA to solve QUBO problem; *XYmixer [39]:* hard constraints and is proved to be an HWP gate with the Hamiltonian as $\mathbf{H}_{XY} = \sigma^x \otimes \sigma^x + \sigma^y \otimes \sigma^y$.

We apply soft constraints to the above baselines by adding a penalty term in the Hamiltonian with $\mathbf{X}$ as the permutation matrix from the final state:

$$C_{Penalty} = \sum_{i=0}^{m-1} \Big( \sum_{j=0}^{m-1} \mathbf{X}_{ij} - 1 \Big)^2 + \sum_{j=0}^{m-1} \Big( \sum_{i=0}^{m-1} \mathbf{X}_{ij} - 1 \Big)^2 \tag{18}$$

$$\mathcal{L}_{QAP-soft} = \langle \psi | \mathbf{H}_{QAP} | \psi \rangle + \alpha \times C_{Panelty}, \tag{19}$$

where $| \psi \rangle$ is the final state and $\alpha$ is the penalty coefficient. Moreover, we adopt XYmixer as a HWP gate in the proposed hard constraint paradigm to see the performance between different HWP gates.

**Evaluation Metric:** To better illustrate the performance difference across methods, we utilize the approximation ratio $\eta$ as the evaluation metric, defined as:

$$\eta = \frac{\mathcal{L}_{max} - \mathcal{L}_{QAP}}{\mathcal{L}_{max} - \mathcal{L}_{min}}, \tag{20}$$

where $\mathcal{L}_{max}$ and $\mathcal{L}_{min}$ denote the maximum and minimum loss for in-constraint states. For an infeasible state, the loss $\mathcal{L}_{QAP}$ is equal to $\mathcal{L}_{max}$, so the approximation ratio $\eta = 0$. For an in-constraint state, $\eta$ is a value between $0$ and $1$, and it evaluates the overall performance. Apart from the approximation ratio, we also evaluate the methods based on the probability of obtaining the optimal solution denoted as $p_{optimal}$. We utilize both metrics to avoid the situation that we converge to a second-best solution which yields high $\eta$ and low $p_{optimal}$.

**Hyperparameter Setting:** In this section, we will discuss two crucial hyperparameters namely the penalty weight $\alpha$ for soft constraints and the number of parity checks $T$ for hard constraints. We first analyze the number of parity checks required in our model. This experiment is conducted with $L \times T$ remains to be a constant so that adding parity checks will not increase the number of parameters. From Fig. 5, we can see that the probability of obtaining feasible states from the last parity check declines slowly, but the probability of obtaining an optimal solution requires a specific number of parity checks. Moreover, parity checks with excessively small intervals might cause the in-constraint states to be trapped at the initial state as the HWP layers in between two parity checks are not able to transfer the initial state to other feasible states. Thus, we set the number of parity checks $T = 4$ to balance the circuit depth and the quality of results. (Details analysis for penalty weight $\alpha$ for soft constraints are in Apx. F.2)

**Results on simulators:** The results are shown in Tab. 3. We provide the dimension of the space in which these approaches operate. When utilizing the soft constraints, HWP ansatz can reduce the dimension from $2^n$ to $\binom{n}{m}$. However, when we use hard constraints, the dimension before the parity check is $m^m$ and it further decreases to $m!$ which is exactly the problem complexity of QAP. Considering the space of those methods, we are unable to provide results for baselines with soft constraints for $m = 5$ (25 qubits) and $m = 6$ (36 qubits).

All the results for soft constraints are conducted regardless of the physical qubit topology. HEA, XYmixer-NN, and NBS-NN satisfy the nearest neighbor topology while QAOA, XYmixer-FC, and NBS-FC require full connectivity of the qubits. We set the penalty $\alpha = 1$ for all the soft constrained baselines based on the analysis in Fig. 5. The results show that: (1) HWP ansatz with soft-constraints are generally better than QAOA and HEA; (2) the dimension of DLA of HWP ansatz matters to the results see Tab. 4; (3) FC leads to better exploration of the subspace and will lead to more out-of-constrained states, which indicates lower $\eta$ when $\alpha$ is set the same as NN.

While soft constraints often face difficulties in identifying optimal solutions at larger scales, hard constraints consistently demonstrate notable effectiveness and robustness. For the hard constrained methods, we show results with NN connectivity only since our method is qubit topology

Table 4: DLA dimension when $n = m^2$.

| NBS | | XYmixer | |
| NN | FC | NN | FC |
| --- | --- | --- | --- |
| $\binom{n}{m}^2$ | $\binom{n}{m}^2$ | $\begin{array}{l} n^2 - 1 \quad n \text{ is odd} \\ \frac{1}{2}n(n-1) \; n \text{ is even} \end{array}$ | $\binom{n}{m}^2 - 1$ |

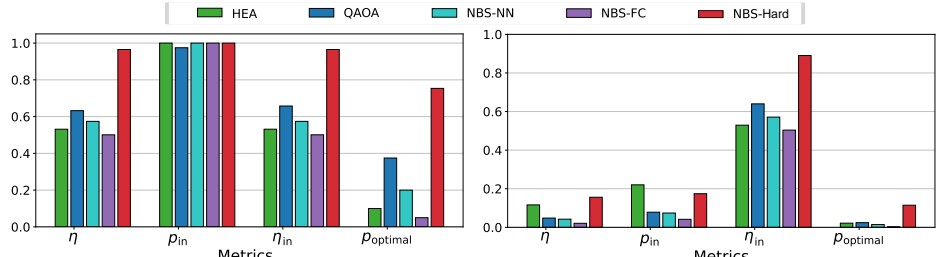

Figure 3: Results for QAP on quantum processor. The upper one is results on noiseless simulator as a standard, the lower one is on superconducting quantum processor, under the same parameter setting.

oriented. XYmixer can obtain a relatively high approximation ratio but with a rapid decrease in the probability of obtaining the optimal solution as $m$ increases. The results verify the capability and efficiency of our hard constrained method as well as the expressivity of the proposed NBS gate.

### 3.4 QAP on Superconducting quantum processors

We further conduct the experiment with a superconducting quantum processor. The 12 qubits (see Fig. 4) are chosen from a 66-qubit superconducting quantum processor. The processor has qubits lying on a 2D lattice, and the qubits are capacitively coupled to their four nearest neighbors. Detailed information about this processor is listed in Sec.D.2. None of the experiments on the quantum processor involve quantum error mitigation methods to post-process data. Considering the qubit quality and coupling strength, here we only conduct experiments for the case of $m = 3$ for QAP, as a primary verification of the feasibility of executing the algorithm on quantum processors. Detailed hyperparameter setting see Apx. F.1

**Evaluation Metric:** Apart from the approximation ratio $\eta$ and the probability of obtaining the optimal solution $p_{optimal}$ used for the simulator, we introduce two more metrics to analyze the performance on the quantum processor. The first one is the approximation ratio for the in-constraint solution denoted as $\eta_{in}$. Since all the infeasible solutions are counted as 0 in $\eta$, $\eta_{in}$ can be seen as the true approximation ratio. Thus, it is very important to include the probability $p_{in}$ of obtaining feasible solutions from all the output solutions as the second metric. Since the circuit error will greatly infect the solutions and $\eta$ will become very small, $\eta_{in}$ can enlarge the difference when conducting experiments on the quantum processor.

**Results:** The numerical results on the quantum processor are illustrated in Fig. 3. Quantum noise greatly affects the results, although we only utilize twelve qubits. QAOA and HEA demonstrate better performance with very few parameters and shallow circuits. All the methods requiring full connectivity will include SWAP gates during compilation, which leads to worse performance on the quantum processor. Our method consistently outperforms other soft constrained methods over all the metrics and further enlarges the overhead when executed on the quantum processor, except for HEA on $p_{in}$ since HEA is extremely shallow and it may perform better on quantum processor some time. We believe the results will be much better if deeper circuits and more two-qubit gates are allowed. Notice that the parity checks on the columns are also able to correct the bit flip errors on the column. This may be another reason why we can achieve better performance with noise.

## 4   Conclusion and Limitations

In this paper, we first demonstrate the efficiency of HWP ansatz on ground state energy estimation problem and explain why HWP ansatz is a perfect testbed for parity check. Results on simulator with noise verify that parity check is a powerful error mitigation method for HWP ansatz. We then propose a novel method to utilize parity check as projective measurement to enforce further hard constraints for HWP ansatz. Intensive experimental results on QAP on both simulator and superconducting quantum processor illustrate the superior performance against peer VQA methods relying on soft constraints. To conclude, we provide detailed evidence in this paper to show that the combination of HWP ansatz and parity check is among the most promising candidates to demonstrate quantum advantages in the NISQ era to solve realistic problems.

**Limitation and Future Work:** We are aware that quantum algorithms might not exhibit any advantage (at least currently) under inevitable quantum noise and the extremely small problem size. However, we believe this paper still break through the upper limit of existing constrained quantum

algorithms for CO. We will further study the performance of our methods on inequality constraints in future. At the moment, our work does not have any negative societal impacts.

## 5 Acknowledgement

The authors would like to express their gratitude to Zhiyuan College, Shanghai Jiao Tong University and Zhiyuan Future Scholar Program (Grants No. ZIRC2024-07).

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

# A  Related Work

In this section, we first briefly review the related works of this paper. A typical way for quantum computers to solve the optimization problem is to transform the optimization problem into a Quadratic Unconstrained Binary Optimization (QUBO) problem and then model the QUBO problem using the Ising model. The Ising model can be solved with variational quantum algorithms (VQAs) with QAOA [27] as the most famous one. However, this paradigm is not natively designed to deal with constraints. Thus, we need to develop special strategies for constrained optimization problems to be solved on quantum computers.

Existing approaches tackle constraints either by adding soft constraints to the Hamiltonian [76, 23, 20], or hard constraints to the quantum circuit [39, 30, 61]. Soft constraints are easy to realize but are hindered by the problem of balancing the objective and constraints [80], which makes hard constraints a better choice in general. However, hard constraints are quite hard to enforce on the quantum circuit since it is impossible to restrict the quantum state to a smaller subspace only by unitary transformations. Different from designing hard constrained ansatz from mining in all the gates, the very recent work [78] proposed a pipeline to analyzing an HWP ansatz. However, they have not yet figured out a proper way to estimate the dimension of the Dynamic Lie Algebra (DLA) of different HWP gates.

QAP, as one of the most significant NP-hard CO problems [68], is defined as finding a minimum cost allocation of facilities to locations, with the costs being the sum of all possible distance-flow products. There have been only very few quantum algorithms for QAP due to the constraints. [71, 49, 4, 21] focus on using soft constraints to solve QAP and [80] utilize QNN to learn from data oriented QAP with constraints are enforced by classical Sinkhorn layer.

Finally we also briefly mention the develop in classic machine learning for solving combinatorial optimization [8], especially those with hard constraints, whereby the constrained are mostly incorporated by penalty term [47]. While some exceptions [74, 75] design specific neural layers to accommodate the permutation constraint or others. By contrast, in this paper, we focus on the quantum realm.

## A.1  LDPC

Classical LDPC codes, first proposed by Gallager in 1962 [31], are binary linear codes defined by a paritycheck matrix $H$. These codes have seen renewed interest due to their low bit error rates under fixed signal-to-noise ratios, particularly with the development of turbo codes and iterative decoding techniques. The defining properties of LDPC codes are: 1) Each row contains $\rho$ ones. 2) Each column contains $\gamma$ ones. 3) The number of ones in common between any 2 columns, denoted $\lambda$, is at most 1. 4) Both $\rho$ and $\gamma$ are small relative to the code length.

Quantum LDPC codes extend the principles of classical LDPC codes into the quantum domain, functioning as stabilizer codes where the stabilizer generators are low-weight operators, thus ensuring efficient error correction in quantum systems. Significant advancements in QLDPC codes began with the introduction of CSS codes [16], which utilized classical codes for quantum error correction. MacKay et al. [56] pioneered sparse graph-based QLDPC codes, enhancing error correction efficiency. Hagiwara and Imai [40] developed quantum quasi-cyclic LDPC codes, which simplified encoding and decoding processes. Camera et al. Gottesman [35] demonstrated that QLDPC codes could significantly reduce the overhead required for fault-tolerant quantum computing. Recent advancements include work by Baspin and Krishna [6], who explored connectivity constraints and provided bounds on code distance and dimension. Panteleev and Kalachev [63] demonstrated the existence of asymptotically good QLDPC codes by leveraging the lifted product construction over non-abelian groups. Additionally, Gu et al. [37]showed that quantum Tanner codes enable single-shot error correction, crucial for practical fault-tolerant quantum computing. LDPC codes are widely regarded as the most promising approach for realizing quantum error correction. However, there is currently no clear evidence of their application on real-world quantum ansatz [12].

## A.2  Constrained Quantum Optimization

Constrained Optimization is one of the most focused applications in the quantum computing domain. For unconstrained optimization problems, quantum processors have established methods, typically involving the transformation of optimization problems into QUBO problems, subsequently reformulated as Ising models, and ultimately solved using Variational Quantum Algorithms. Thus, the key to

solving Constrained Optimization lies in handling constraints, which are divided into soft constraints that encode penalty terms into the Hamiltonian, and hard constraints that restrict the evolutionary space using quantum circuits.

**Soft Constraints**

The soft constraint method, currently the most commonly employed, incorporates penalty terms into the objective function, offering adaptability to a wide range of constraints and high versatility. However, the quality of the solution is highly sensitive to the balance between the objective and constraints [76]. Under soft constraints, quantum circuits may return all infeasible and feasible solutions, with penalty terms ensuring a higher probability of feasible solutions appearing in the outcome. This approach requires the algorithm to explore the entire $B^n$ space, where $B^n$ represents the binary string typically equivalent to the number of qubits. Compared to the feasible space of solutions, this represents a significantly larger search domain, leading to a decrease in the feasibility rate and search efficiency of solutions.There is extensive research on methods employing soft constraints.For instance, the Constrained Binary Model solver (CQM) developed by D-Wave[23] incorporates a series of linear constraints and inequality constraints.[20]discusses various approaches to encoding constrained optimization and constraint satisfaction problems into QUBO issues, specifically targeting problems that involve at most one constraint.[26] utilizes the quantum phase estimation method, which enables the search for an item in a sorted or unsorted database. However, it does not guarantee a feasible solution.[11] primarily addresses the constraints of quantum annealing hardware and proposes new algorithms for mapping Boolean constraint satisfaction problems onto quantum annealing hardware.

**Hard Constraints**

Hard constraints involve restricting quantum evolution within a in-constraint subspace, ensuring that all obtained solutions are feasible.Hard constraints involve restricting quantum evolution within an in-constraint subspace, ensuring that all obtained solutions are feasible. However, most hard constraint methods are still only capable of solving specific and relatively simpler forms of problems. An example of this approach is the Quantum Alternating Operator Ansatz(QAOA-c) proposed by [39]. This method is an extension of QAOA and allows for alternation between families of unitary operators with general parameterizations. The challenge with QAOA-c lies in the complexity of implementing constraint-preserving mixers and the necessity to introduce a large number of auxiliary qubits, leading to a lack of universality. Under current hardware conditions, preparing a uniform superposition of constrained states also presents a significant challenge.[30]Additionally, this approach suffers from limited flexibility in its application.There has been much research related to QAOA-c. For instance, [61] executed the Quantum Alternating Operator Ansatz algorithm on a trapped-ion quantum computer, utilizing Hamming-weight-preserving XYmixer circuits to restrict quantum evolution to the in-constraint subspace. [69] proposed the dynamic quantum variational ansatz, which dynamically adapts to ensure maximum utilization of a fixed allocation of quantum resources.

In addition to QAOA-c and its derivative algorithms, there are also works on other hard constraint methods. For example, [78] analyzed the capability, expressivity, and trainability of Hamiltonian-weight preserving ansatz and verified these theoretical results on the unitary approximation problem.[42] restricts quantum evolution to the constrained subspace through repeated projective measurements. However, this method has not been effectively validated on many quantum gates and is particularly sensitive to the number of measurements.

### A.3 Quadratic Assignment Problem

The Quadratic Assignment Problem (QAP) is defined as finding a minimum cost allocation of facilities to locations, with the costs being the sum of all possible distance-flow products. The QAP is one of the most significant combinatorial optimization problems, and it has been proven to be NP-hard [68].

**Classic Algorithms**

The predominant methodologies for solving this encompass the exact, heuristic, and metaheuristic approaches. In the following sections, we will discuss these methodologies and present typical techniques within each category.

Obtaining exact solutions for QAP is extremely challenging, and current methodologies are only capable of achieving global optimal solutions for small-scale QAP instances. Existing approaches include branch-and-bound, cutting planes, or combinations of these methods. The branch-and-

bound algorithm[57, 2] commences with a heuristic-derived initial feasible solution, setting the upper bound, and then segments the problem into lower-bounded sub-problems. [1]The cutting planes method[7] employs Mixed Integer Linear Programming formulations but is characterized by extended computational times. The branch-and-cut algorithm[62] amalgamates the aforementioned methodologies, offering an advantage over cutting planes as the cuts are associated with the polytope's facets, enabling swifter convergence.Additionally, dynamic programming[55] is another method[19] for obtaining exact solutions , yet it is incapable of running in polynomial time.

Heuristic methods, although unable to guarantee optimal solutions, can yield reasonable solutions within a shorter time frame.Numerous heuristic methods have been proposed for addressing QAP. For instance, [32] introduced the constructive method. [14] applied enumerative methods, and improvement methods were utilized by [15].

Metaheuristics represent a more universal paradigm, essential to which are a priori strategies adapted to the problem structure. Metaheuristics can be categorized into single-based solutions and population-based solutions.[1] For instance, Simulated Annealing [51] is a type of single-based solution, while Genetic Algorithms [43] fall under population-based solutions. Other methods include Scatter Search[33] and Ant Colony Optimization[25].

**Quantum Algorithms**

Due to the complexity of QAP, there are currently few quantum solvers specifically designed for QAP. Most of the existing solvers predominantly employ the method of soft constraints.[49, 50]utilize traditional soft constraint encoding, experiments were conducted on IBM's quantum devices using the VQE and Quantum QAOA solvers, respectively.[54]focuses on solving sparse Quadratic Assignment Problems (QAP), the approach involves transforming quadratic terms generated by penalty items into linear terms, followed by a post-processing step to derive feasible solutions. [80]presents the construction of a novel quantum neural network, whereby feasible solutions are obtained through classical post-processing. [4]focuses on extending the method to bi-objective QAP expressed in the form of Quadratic Unconstrained Binary Optimization (QUBO), without altering the original soft constraints.[71, 21]employ quantum annealing to solve problems, similarly incorporating constraints as penalty terms.[9]proposes an approach of updating cycles as a substitute for constraints in QAP. However, the energy calculation for cycles is only feasible for certain specialized versions of QAP, such as the three-dimensional shape correspondence problems highlighted in the paper.

## B  Preliminaries

We will first introduce one of the most common type of symmetry-preserving ansatz, namely the HWP ansatz, and basic ideas of parity check in the quantum circuits. Then we will provide details about quantum computing and machine learning.

### B.1  Hamming Weight Preserving Ansatz

Now we give a brief review of the Hamming Weight Preserving ansatz. The HWP ansatz aims to solve a fundamental physical symmetry which is the number of spin-ups in the quantum states. For a $n$-qubit quantum system with the number of $|1\rangle$s in the quantum states is $k$, we say that this problem is in an HWP subspace with the dimension of the subspace as $d_k = \binom{n}{k}$. The work [78] provides a detailed definition of the HWP ansatz as well as a thorough analysis of the expressivity, capability, and trainability of the ansatz.

For two-qubit HWP situation, all four basis states $\{|00\rangle, |01\rangle, |10\rangle, |11\rangle\}$ have three Hamming weights: 0, 1, and 2. Thus, a two-qubit HWP gate should only operate on basis states $|01\rangle$ and $|10\rangle$. The general form of the two-qubit HWP gates is:

$$\mathbf{H}_{HW} = \begin{pmatrix} 0 & 0 & 0 & 0 \\ 0 & a & b & 0 \\ 0 & \bar{b} & c & 0 \\ 0 & 0 & 0 & 0 \end{pmatrix}, \tag{21}$$

where $a, c \in \mathbb{R}$, $b \in \mathbb{C}$, and $\bar{b}$ denotes the conjugate of $b$. $\mathbf{H}_{HW}$ is a Hermitian matrix that satisfies $\mathbf{H}_{HW}^{\dagger} = \mathbf{H}_{HW}$. Taking the Hermitian matrices of the allowed gates as generators, we have a set of generators $\mathcal{G} = \{\mathbf{H}_p\}_{p=1}^{P}$, we can define the Dynamical Lie Algebra (DLA):

$$\mathfrak{g} = span \langle i\mathbf{H}_1, i\mathbf{H}_2, \cdots, i\mathbf{H}_P \rangle_{Lie}, \tag{22}$$

where $\langle \cdot \rangle_{Lie}$ denotes the Lie closure. We then repeatedly take the commutators of the elements in the generator set. For a $N$-dimensional quantum system, we have the following important theorem [66].

**Theorem B.1.** *[66] A necessary and sufficient condition for complete controllability of a $N$-dimensional quantum system $\hat{\mathbf{H}}$ is that the dimension of the DLA $\mathfrak{g}$ is $N^2$ where $N$ is the dimension of quantum system.*

## B.2 Quantum Chemistry

We provide the definition of annihilation operator and creation operator in second quantization.

**Definition B.2.**
$$a_i^\dagger = \begin{pmatrix} 0 & 1 \\ 0 & 0 \end{pmatrix}, \quad a_j = \begin{pmatrix} 0 & 0 \\ 1 & 0 \end{pmatrix}, \tag{23}$$

where $a_j$ denotes the annihilation operator on qubit $j$ and $a_i^\dagger$ denotes the creation operator on qubit $i$.

## B.3 Quantum Computing

In quantum computing, 'qubit' (abbreviation of 'quantum bit') is a key concept which is similar to a classical bit with a binary state. The two possible states for a qubit are the state $|0\rangle$ and $|1\rangle$, which correspond to the state 0 and 1 for a classical bit respectively. We refer the readers to the textbook [60] for comprehension of quantum information and quantum computing. Here we give a brief introduction to the background.

A quantum state is commonly denoted in bracket notation. It is also common to form a linear combination of states, which we call a superposition: $|\psi\rangle = \alpha|0\rangle + \beta|1\rangle$. Formally, a quantum system on $n$ qubits is an $n$-fold tensor product Hilbert space $\mathcal{H} = (\mathbb{C}^2)^{\otimes d}$ with dimension $2^d$. For any $|\psi\rangle \in \mathcal{H}$, the conjugate transpose $\langle\psi| = |\psi\rangle^\dagger$. The inner product $\langle\psi|\psi\rangle = ||\psi||_2^2$ denotes the square of the 2-norm of $\psi$. The outer product $|\psi\rangle\langle\psi|$ is a rank 2 tensor. Computational basis states are given by $|0\rangle = (1,0)$, and $|1\rangle = (0,1)$. The composite basis states are defined by e.g. $|01\rangle = |0\rangle \otimes |1\rangle = (0,1,0,0)$.

## B.4 Quantum Machine Learning

[17] proposed the concept of Variational Quantum Algorithms (VQA), which leverages quantum advantages to solve machine learning problems on a near-term quantum device. Then, Parameterized Quantum Circuits (PQC) are the concrete implementation of certain VQA. For each qubit we have rotation operator $Rx(\boldsymbol{\theta})$ which rotate through angle $\boldsymbol{\theta}$ (radias) around the $x$-axis. A PQC is mainly composed of $Rx(\boldsymbol{\theta})$, $Ry(\boldsymbol{\theta})$ and $Rz(\boldsymbol{\theta})$ with $\boldsymbol{\theta}$ as the parameters. The parameters $\boldsymbol{\theta}$ are updated by a classical optimizer to minimize the loss function $\mathcal{L}(\boldsymbol{\theta})$ which evaluates the dissimilarity between the output of PQC and the target result. The derivative of the $i$-th parameter $\boldsymbol{\theta}(i)$ can be computed by using the shifting technique proposed by [59]. It requires running the whole circuit twice but with shifting $\boldsymbol{\theta}(i)$ to $\boldsymbol{\theta}(i) + \pi/2$ and $\boldsymbol{\theta}(i) - \pi/2$

$$\begin{aligned}
\frac{\partial\mathcal{L}(\boldsymbol{\theta})}{\partial\boldsymbol{\theta}(i)} =& \frac{1}{2} \times \left( \mathcal{L}\left(\boldsymbol{\theta}(1), \cdots, \boldsymbol{\theta}(i) + \frac{\pi}{2}, \cdots\right) \right. \\
& \left. - \mathcal{L}\left(\boldsymbol{\theta}(1), \cdots, \boldsymbol{\theta}(i) - \frac{\pi}{2}, \cdots\right) \right)
\end{aligned} \tag{24}$$

# C  Proof of Theorem 2.3

We first restate the theorem:

**Theorem.** *An ansatz $\mathbf{U}(\boldsymbol{\theta})$ can solve the ground state energy estimation problem without truncation if the ansatz is universal under the $d_k$-dimensional HWP subspace.*

*Proof.* Firstly, the ground state energy estimation problem has the physical symmetry that the number of total spins and total particles will not change during the evolution. So the final state will have exactly the same number of $|1\rangle$s as the initial state (usually the Hatree-Fock state). Thus, the ground state energy estimation problem can be solved within a HWP subspace with the dimension as $d_k = \binom{n}{k}$.

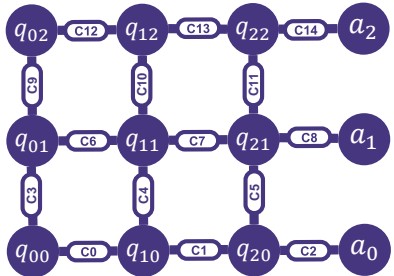
Figure 4: Selected qubits on the superconducting quantum processor

Secondly, an ansatz $\mathbf{U}(\boldsymbol{\theta})$ is universal if and only if the reachable unitary matrices from arbitrary parameters $\boldsymbol{\theta}$ satisfies:

$$\{\mathbf{U}(\boldsymbol{\theta})\}_{\boldsymbol{\theta}} = \mathcal{SU}(d_k), \tag{25}$$

where $\{\mathbf{U}(\boldsymbol{\theta})\}_{\boldsymbol{\theta}}$ denotes the reachable unitary matrices from arbitrary parameters $\boldsymbol{\theta}$ and $\mathcal{SU}(d_k)$ denotes the super unitary group with dimension as $d_k$. Thus, we can approximate any unitary matrix within the HWP subspace with arbitrary precision by optimizing the parameters in the universal ansatz.

Back to the ground state energy estimation problem. Since we are finding the optimal eigenstate with the smallest eigenvalue, we can take this problem as a state preparation problem. If we have a universal ansatz in HWP subspace, then we can reach arbitrary state in the HWP subspace with any legit initial state. Therefore, we conclude that the ansatz $\mathbf{U}(\boldsymbol{\theta})$ can solve the ground state energy estimation problem without truncation if the ansatz is universal under the $d_k$-dimensional HWP subspace. □

# D Implementation Detail

All the numerical simulations are performed on a machine with 190GB memory, one physical CPU with 32 cores AMD Ryzen Threadripper 3970X CPU, and 5 GPUs (Nvidia GeForce RTX 3090). We implement a Python quantum simulator so that we are able to simulate a quantum system for our method with up to 42 qubits ($m = 6$ with ancilla qubits). Implementation details are shown in Appendix D.1. The results on the quantum processor are conducted on a superconducting quantum processor with detailed information shown in Appendix D.2.

## D.1 Implementation Detail for the Classical Simulator

**self-built simulator** Since the quantum circuit of the proposed hard constrained method is executed in the $m^m$ dimensional subspace, we utilize a implementation trick to reduce the dimension of the simulation. For a traditional quantum simulator based on quantum states, the space we need to store a quantum state is $2^{m \times m}$. However, we can utilize a mapping function to map the corresponding state to the $m^m$ dimensional vector. For each two-qubit gates on the row, the gate is operating on a $m * 1$ dimensional quantum state, which indicates the gate is a $m \times m$ unitary operator. Therefore, the time complexity of applying a quantum gate on the state is $\mathcal{O}(m^{m-1}m^2)$, which is much faster than $\mathcal{O}(2^{m \times m})$. Moreover, to simulate the projective measurement, we simulate the results on the shot basis, enabling us to simulate the in-circuit measurement.

## D.2 Implementation Detail for Superconducting Quantum Processor

In the experiment, we utilized a 12-qubit Superconducting Quantum Processor with an intercoupled topology as shown in the Fig. 4.Tab. 5 presents the performance information regarding Single-Qubit Operations, while Tab. 6 presents the fidelity of the controlled-Z gate between two qubits.

| Qubit | $q_{00}$ | $q_{01}$ | $q_{02}$ | $q_{10}$ | $q_{11}$ | $q_{12}$ | $q_{20}$ | $q_{21}$ | $q_{22}$ | $a_0$ | $a_1$ | $a_2$ |
|---|---|---|---|---|---|---|---|---|---|---|---|---|
| $\omega/2\pi$(GHz) | 4.8661 | 4.6814 | 4.6302 | 4.9031 | 4.7910 | 4.7097 | 4.8504 | 4.9323 | 4.8387 | 4.709 | 4.5704 | 4.7247 |
| $T_1(\mu s)$ | 26.298 | 33.374 | 28.819 | 28.239 | 20.626 | 36.092 | 21.695 | 24.781 | 28.757 | 31.169 | 37.170 | 31.313 |
| $T_2^*(\mu s)$ | 2.8271 | 2.6789 | 2.2952 | 5.8565 | 2.9713 | 3.1998 | 3.8482 | 4.8506 | 1.873 | 4.459 | 1.6298 | 1.1544 |
| $F_0$ | 0.9414 | 0.9222 | 0.9168 | 0.9242 | 0.9268 | 0.9478 | 0.9400 | 0.9206 | 0.9592 | 0.9364 | 0.9604 | 0.9240 |
| $F_1$ | 0.8732 | 0.8022 | 0.8690 | 0.8062 | 0.7378 | 0.8686 | 0.8430 | 0.8166 | 0.8484 | 0.703 | 0.8404 | 0.8276 |
| $F_G$ | 0.9990 | 0.9993 | 0.9992 | 0.9993 | 0.9992 | 0.9994 | 0.9991 | 0.9993 | 0.9989 | 0.9993 | 0.9987 | 0.9989 |

Table 5: Performance of Single-Qubit Operations. $\omega$ is the working frequency. Coherence times T1 and $T2^*$ representing the energy relaxation time and dephasing time. The term $F_0$ ($F_1$) represents the readout fidelity, specifically denoting the probability of accurately measuring the qubit state in $|0\rangle$ ($|1\rangle$) after it has been successfully initialized in the $|0\rangle$ ($|1\rangle$) state. $F_G$ denotes the gate fidelity of native gates.

| Qubit | $q_{00}$ | $q_{01}$ | $q_{02}$ | $q_{10}$ | $q_{11}$ | $q_{12}$ | $q_{20}$ | $q_{21}$ | $q_{22}$ | $a_0$ | $a_1$ | $a_2$ |
|---|---|---|---|---|---|---|---|---|---|---|---|---|
| $q_{00}$ | - | 0.9839 | - | 0.9827 | - | - | - | - | - | - | - | - |
| $q_{01}$ | 0.9798 | - | 0.9806 | - | 0.9778 | - | - | - | - | - | - | - |
| $q_{02}$ | - | 0.9806 | - | - | - | 0.9798 | - | - | - | - | - | - |
| $q_{10}$ | 0.9827 | - | - | - | 0.9868 | - | 0.9867 | - | - | - | - | - |
| $q_{11}$ | - | 0.9778 | - | 0.9868 | - | 0.9852 | - | 0.9840 | - | - | - | - |
| $q_{12}$ | - | - | 0.9798 | - | 0.9852 | - | - | - | 0.9873 | - | - | - |
| $q_{20}$ | - | - | - | 0.9867 | - | - | - | 0.9880 | - | 0.9822 | - | - |
| $q_{21}$ | - | - | - | - | 0.9840 | - | 0.9880 | - | - | - | 0.9783 | - |
| $q_{22}$ | - | - | - | - | - | 0.9873 | - | - | - | - | - | 0.9791 |
| $a_0$ | - | - | - | - | - | - | 0.9822 | - | - | - | - | - |
| $a_1$ | - | - | - | - | - | - | - | 0.9783 | - | - | - | - |
| $a_2$ | - | - | - | - | - | - | - | - | 0.9791 | - | - | - |

Table 6: The controlled-Z gate fidelity between qubit pairs, which is symmetric along the main diagonal due to the undirected nature of qubit coupling.

# E  Further Experimental Results for Parity Check as Error Mitigation

## E.1  Hyperparameter Setting

Here we provide the noise levels we used in the experiments. We set the depolarizing error for single qubit as $1.6 \times 10^{-3}$ and for two-qubit gates as $6.4 \times 10^{-3}$. The bit-flip and phase-flip error for each gate are set as $5 \times 10^{-3}$. We also set a $1 \times 10^{-2}$ readout error for obtaining $|1\rangle$ given $|0\rangle$, and $5 \times 10^{-2}$ vise versa. To simulate the computing way on quantum computers, we set the number of shots per Pauli string in the measurement stage at 5000, which makes at least 500,000 shots in total for LiH and $H_2O$ since they all have more than 100 Pauli strings.

## E.2  Sensitivity Analysis on the Number of Parity Checks

Now we discuss the impact of the number of parity check layers. It is clear that more parity checks can reduce the probability of errors but it also brings in more CNOT gates which can also hinder the results. Thus, it is crucial to seek equilibrium in the number of parity checks in the NISQ era. From the results in Tab. 7, we conclude that applying a parity check for every 6 HWP gates can achieve the best performance. Thus, we insert 2 parity check blocks for $H_2$ and 10 blocks for LiH and $H_2O$. We omit the analysis on $H_2$ since the ansatz is very shallow for $H_2$. Consider that we have 63 HWP gates in the 8-qubit ansatz, we set the number of parity checks as 6, 8, 10, 12, 15, which is a parity check per 10, 8, 6, 5, 4 gates respectively. The results illustrate that a parity check for every 6 HWP gates can have the best performance considering the additional CNOT gates brought in by parity checks.

## E.3  Ansatz statistics

Detailed statistical information for both UCC ansatze and the proposed HWP ansatz in Tab. 8. We provide the number of gates, number of CNOT gates and the number of parameters to show the

| energy error | num_PC=6 | num_PC= 8 | num_PC=10 | num_PC=12 | num_PC=15 |
|---|---|---|---|---|---|
| LiH | 0.657022752 | 0.661989975 | **0.600911213** | 0.65741316 | 0.675546743 |
| $H_2O$ | 0.748342082 | 0.72675715 | **0.688285861** | 0.743041427 | 0.745211306 |

Table 7: Sensitivity analysis on the number of parity checks for the proposed HWP ansatz. The best results are in bold.

| | $H_2$ | | | LiH | | | $H_2O$ | | | |
|---|---|---|---|---|---|---|---|---|---|---|
| | UCCS | UCCSD | ours | UCCS | UCCSD | ours | UCCS | UCCSD | UCCSDT | ours |
| num_gates | 32 | 107 | 84 | 108 | 1067 | 441 | 144 | 2002 | 5871 | 441 |
| num_CNOT | 8 | 56 | 36 | 48 | 732 | 189 | 64 | 1376 | 4304 | 189 |
| num_params | 2 | 3 | 12 | 6 | 15 | 63 | 15 | 26 | 34 | 63 |

Table 8: Statistics for UCC and proposed HWP ansatze.

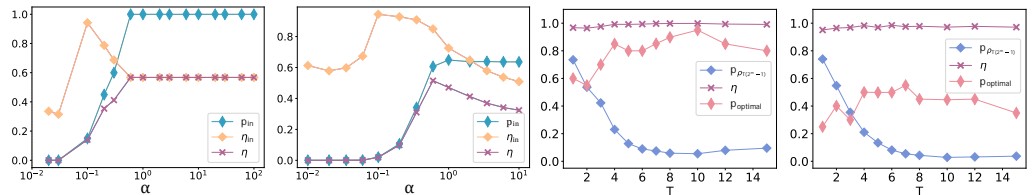

Figure 5: From left to Right. (a) and (b) Sensitivity study on penalty weight $\alpha$: (a) XYmixer-NN with $m = 3$; (b) XYmixer-FC with $m = 4$. (c) and (d) Sensitivity study on the number of parity checks $T$. (c) NBS-NN with $m = 5$; (d) NBS-NN with $m = 6$.

difference between the baseline methods and ours. It is shown that the proposed HWP ansatz can achieve precise energy results with much shallower circuit than UCCSD and UCCSDT.

## F  Further Experimental Results for Parity Check as Further Constraints

### F.1  Hyperparameter Setting

**QAP on simulator:** The layer of XYmixer-FC, NBS-FC, HEA, and QAOA are set to $2 \times m$ and XYmixer-NN, NBS-NN, XYmixer-hard, and NBS-hard have the same number of parameters as NBS-FC. Learning rate is set to 0.05 and the number of iterations is 1000.

**QAP on Superconducting Quantum Processors:** Due to limited coherence time and noise, the parameter setting is a bit different from that on the simulator. We are only able to apply one layer of NBS-FC considering the number of CNOTs, so the rest of the methods are set with the same amount of parameters as one layer of NBS-FC. As QAOA has a parameter-sharing strategy, we set its number of layers as 2. For NBS-Hard, we set $T = 2$. Each circuit is measured with 20K shots.

### F.2  Sensitivity Analysis for QAP

The penalty weight is very sensitive and may vary significantly with different models and different problem sizes. We set $\alpha = 1$ as a constant based on the sensitivity results on XYmixer since it is impossible to fine-tune $\alpha$ for each instance. As illustrated in Fig. 5, the approximation ratio $\eta$ achieves a peak at $\alpha = 1$ (figures for other scenarios are listed in Fig. 6). We provide $\eta$ as the approximation ratio for all states, $\eta_{in}$ as the approximation ratio for feasible states, and $p_{in}$ as the probability of obtaining feasible states. We observe that smaller $\alpha$ leads to a better in-constraint approximation ratio but a relatively small in-constraint probability. A large $\alpha$ leads to a high $p_{in}$, yet it results in convergence to a random in-constrain state due to the dominance of the penalty term.

### F.3  Simulated Experiments on TSP

**Reduction to QAP:** The complexity of QAP lies in its inclusion of dual relationships, namely the interrelations between locations and the interactions between facilities. When the dual relationship is

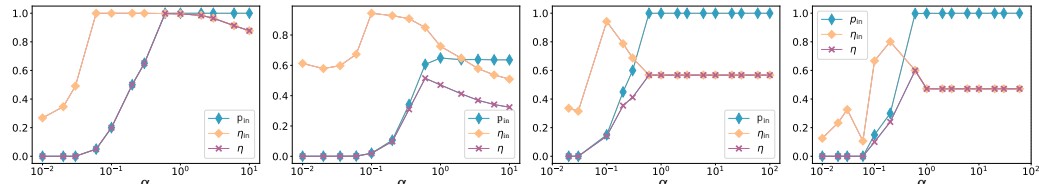

Figure 6: Sensitivity study on penalty weight $\alpha$. From left to right. (a) XYmixer-FC with $m = 3$. (b) XYmixer-FC with $m = 4$. (c) XYmixer-NN with $m = 3$. (d) QAOA with $m = 3$.

Table 9: Results on the simulator for TSP with the best in bold and the best results by soft constraints are underlined.

| CONSTRAINT | METHOD | $dim$ | $\eta$ | | | | $p_{optimal}$ | | | |
|---|---|---|---|---|---|---|---|---|---|---|
| | | | $m=3$ | $m=4$ | $m=5$ | $m=6$ | $m=3$ | $m=4$ | $m=5$ | $m=6$ |
| SOFT | HEA [46] | $2^n$ | 0.7885 | 0.0009 | — | — | 0.6500 | 0.0000 | — | — |
| | QAOA [27] | $2^n$ | 0.7242 | 0.6686 | — | — | 0.4996 | 0.1996 | — | — |
| | XYMIXER-NN [39] | $\binom{n}{m}$ | 0.6857 | 0.7355 | — | — | 0.5497 | 0.3995 | — | — |
| | XYMIXER-FC [39] | $\binom{n}{m}$ | 1.0000 | 0.7096 | — | — | 1.0000 | 0.4531 | — | — |
| | NBS-NN | $\binom{n}{m}$ | 0.9975 | 0.8042 | — | — | 0.9492 | 0.3495 | — | — |
| | NBS-FC | $\binom{n}{m}$ | 1.0000 | 0.5834 | — | — | 1.0000 | 0.1815 | — | — |
| HARD | XYMIXER-NN [39] | $m!$ | **1.0000** | 0.9991 | 0.9920 | 0.9670 | **1.0000** | 0.9500 | 0.8500 | 0.6500 |
| | NBS-NN | $m!$ | **1.0000** | **1.0000** | **0.9933** | **0.9698** | **1.0000** | **1.0000** | **0.9000** | **0.7500** |

simplified to a singular relationship, the problem evolves from a complex assignment decision to a singular sequence or path problem, which is the quest for a Hamiltonian cycle with minimal cost, degenerating QAP into the Traveling Salesman Problem (TSP) formulated as:

$$\min \sum_{k,p} \mathbf{D}_{kp} \sum_i \mathbf{X}_{ik} \mathbf{X}_{(\mathbf{i} \oplus \mathbf{1}) \mathbf{p}}, \tag{26}$$

where $i \oplus 1 = (i + 1) \mod m$. Therefore, TSP can be considered as a special case of QAP under specific simplified assumptions, enabling us to utilize QAP solvers directly for deriving solutions to TSP problems.

**Dataset:** Similar to the QAP dataset, we generate a dataset of random TSP instances, with 100 instances for each size $m = \{3, 4, 5, 6\}$. For each TSP instance, we included a $m \times m$ distance matrix $\mathbf{D}$, which was constructed using randomly generated real points [72].

**Results on Simulator:** The results in Tab. 9 demonstrate a similar pattern as those of QAP. Notice that the TSP problem can be solved by dynamic programming with the algorithm complexity as $\mathcal{O}(n^2 2^n)$ [22], it is theoretically a simpler problem than QAP. This explains why all the methods perform better on TSP. The results further verify the capability and efficiency of the proposed hard constrained method.

