# OpenReview forum: "Rethinking Parity Check Enhanced Symmetry-Preserving Ansatz"
_NeurIPS.cc/2024/Conference — NeurIPS 2024 poster_

### Official Review · Reviewer_kL3u · 2024-06-18

**Soundness:** 4
**Presentation:** 4
**Contribution:** 4
**Rating:** 8
**Confidence:** 5

**Summary:**

This paper designs a novel quantum algorithm that eliminates the need for penalty terms when solving constrained problems. Instead, it constructs a subspace where the states satisfy the constraints, allowing the final solution to be constructed within this subspace. Additionally, the authors use parity check to replace system validation, ensuring that qubits remain unflipped at every step, not just in the final state, thereby enhancing error correction capabilities. The authors demonstrate how this algorithm works on a class of NP-complete combinatorial optimization problems and achieve better results than previous quantum algorithms.

**Strengths:**

**Originality:** The authors propose a quantum machine learning algorithm capable of solving constrained optimization problems without relying on penalty terms.

**Clarity:** The paper clearly defines quantum computing concepts, provides detailed derivations of the main equations involved, and offers a very clear explanation of the quantum system's architecture. The discussion of experimental results is also very thorough.

**Significance:** The quantum algorithm presented in this paper naturally incorporates constraint conditions into the optimization process. Compared to soft constraints, this approach yields more accurate results. This has significant implications for integrating machine learning into quantum computing, offering valuable insights.

**Quality:** The language of the paper is rigorous and clear, with appropriate citations of previous work. The overall approach is straightforward and easy to understand. The paper provides a multi-faceted analysis of the experimental results, validating the algorithm's correctness on simulators and conducting experiments on real quantum computers, further demonstrating feasibility. The theoretical section is also excellent, with detailed derivations of the main equations. The paper excels in both theory and experiment.

**Weaknesses:**

The discussion of the scalability of the proposed algorithm is not sufficient.

**Questions:**

See the comments.

**Limitations:**

yes.

---

> ### Author Rebuttal · Authors · 2024-08-06
>
> Thank you for your thorough review and valuable feedback. We greatly appreciate the time and effort you have invested in evaluating our manuscript. Your insights and suggestions have been instrumental in improving the quality of our work. We are aware that the scalability of the quantum algorithm has attracted considerable interest, and we will provide more detailed derivation and discussion as an extension to Equation 17 in the revision.

---

> > ### Comment · Reviewer_kL3u · 2024-08-14
> >
> > Thanks.

---

### Official Review · Reviewer_6BLB · 2024-07-11

**Soundness:** 3
**Presentation:** 2
**Contribution:** 3
**Rating:** 6
**Confidence:** 1

**Summary:**

Paper introduces a Hamming Weight Preserving ansatz with parity check. The proposed method is tested on quantum chemistry problems and constrained combinatorial optimization problems (eg. quadratic assignment problem). The HWP ansatz can satisfy the hard constraints in QAP. Experiments show that proposed methods outperforms some existing methods for the simulation and superconducting quantum processor.

**Strengths:**

Paper addresses a significant problem of simulation on quantum circuits and applications to quantum chemistry and combinatorial problems. Originality: Paper proposes a novel ansatz using HWP and parity checks, although I am not too familiar with previous work. The quality and clarity of presentation is good, and experiment results are clear.

**Weaknesses:**

A potential weakness may be less background information included in main text of paper, especially for the target audience (ML/neurips).

**Questions:**

Would quantum chemistry ground-state experiment be possible to perform on superconducting processor, and how would methods perform relatively with the quantum noise?

NBS-NN and NBS-FC (if nearest neighbor vs fully connected qubits), these are also commonly abbreviations for Neural network and fully connected (layers). Could authors further describe difference between NBS-NN and NBS-FC? and their difference with NBS-hard?

**Limitations:**

A stated limitation is different results on superconducting processor vs. simulator, based on noise of superconducting processor. Additional stated limitations are related to state of quantum computing such as small problem size and quantum noise.

---

> ### Author Rebuttal · Authors · 2024-08-06
>
> We greatly appreciate the time and effort you have invested in evaluating our paper, and we hope we can clarify your concerns.
>
> > W1: A potential weakness may be less background information included in main text of paper, especially for the target audience (ML/neurips).
>
> Ans: Thanks for your kind advice. We do realize this paper may not be familiar to the general NeurIPS community. We promise to provide a detailed background intro in the revision appendix.
>
> > Q1: Would quantum chemistry ground-state experiment be possible to perform on superconducting processor, and how would methods perform relatively with the quantum noise?
>
> Ans: Ground state preparation is considered one of the most promising problems to demonstrate experimental advantages on recent superconducting processors [1], as evidenced by various preliminary attempts [2, 3, 4]. However, inevitable noise severely affects the performance of current ansätze, prompting the search for new ansätze that are better suited for error mitigation and error correction methods.
>
> > Q2: NBS-NN and NBS-FC (if nearest neighbor vs fully connected qubits), these are also commonly abbreviations for Neural network and fully connected (layers). Could authors further describe difference between NBS-NN and NBS-FC? and their difference with NBS-hard?
>
> Ans: We apologize for any confusion caused by the abbreviations. The terms NBS-NN and NBS-FC refer to nearest-neighbor and fully connected physical qubit connectivity, respectively. On a superconducting quantum processor, physical qubits rely on couplers to achieve entanglement, meaning two-qubit gates can only be applied to qubits directly connected by a coupler. Given the impracticality of connecting each pair of qubits with a coupler, it is not feasible to apply two-qubit gates freely to any pair. Therefore, fully connected (FC) connectivity represents an idealized scenario, whereas studying nearest-neighbor (NN) connectivity is crucial in the Noisy Intermediate-Scale Quantum (NISQ) era, given the current hardware limitations.
>
> In Table 3, NBS-hard denotes the use of parity checks as additional constraints. The simple HWP ansatz can only maintain symmetry along either rows or columns, necessitating extra methods to impose constraints in the other direction. Consequently, NBS-NN and NBS-FC in the soft section of Table 3 represent NBS with corresponding connectivity, where penalties in the Hamiltonian act as constraints. NBS-hard refers to NBS-NN with parity checks as a hard constraint for the other direction.
>
>
>
> [1] The Variational Quantum Eigensolver: A review of methods and best practices. Physics Reports
>
> [2] Hardware-efficient variational quantum eigensolver for small molecules and quantum magnets. Nature
>
> [3] Observing ground-state properties of the Fermi-Hubbard model using a scalable algorithm on a quantum computer. Nature Communications
>
> [4] Experimental quantum computational chemistry with optimized unitary coupled cluster ansatz. Nature Physics

---

### Official Review · Reviewer_K5Kf · 2024-07-12

**Soundness:** 2
**Presentation:** 3
**Contribution:** 2
**Rating:** 5
**Confidence:** 3

**Summary:**

The paper investigates combining the HW-preserving ansatz with qubit topology-aware parity checks to impose hard constraints on quantum circuits for the quantum chemistry and Quadratic Assignment Problem. It includes numerical simulations and experiments on a real quantum device.

**Strengths:**

sufficient numerical and real device experiments.

**Weaknesses:**

comparing to previous work "Rethinking the symmetry-preserving circuits for constrained variational quantum algorithms",  it is difficult to gauge its novelty and impact as the main contribution is incremental with respect to previous results no matter in theoretical or numerical aspects.

**Questions:**

1. as definition of NBS is kind of constant gate, how to train the HWP anstaz constructed by such fixed gates shown in Fig 1.  ? and what is the R in Figure 1 ?
2. comparing to the proposed HWP, the UCCS not only requires shallow circuit but also have out-performance. What is the advantages of the proposed over it?
3. it may be poor scalable of such ansatz used for combinatorial optimization problem, especially in NISQ era. and in line 252, the probability of finding the feasible state will exponentially small with the problem size $m$, how to extract such feasible state without performing exponentially measurement?

**Limitations:**

no potential negative societal impact of their work.

---

> ### Author Rebuttal · Authors · 2024-08-06
>
> Thank you for your thorough review and valuable feedback, including the comparison of previous work [1]. We would like to highlight the differences between these two papers and promise to add a remark in the related work section to distinguish them.
>
> In [1], the authors proposed methods to analyze the expressivity and trainability of the HWP ansatz. They also proposed BS gate to conduct experiments with the Hamiltonian as:
> $$
> H_{BS}=\left(\begin{array}{cccc}
>         0 & 0 & 0 & 0\\\\
>         0 & \frac{1}{2} & \frac{1+\text{i}}{2\sqrt{2}} & 0\\\\
>         0 & \frac{1-\text{i}}{2\sqrt{2}} & \frac{1}{2}& 0\\\\
>         0 & 0 & 0 & 0
>     \end{array}\right),\quad H_{NBS}=\left(\begin{array}{cccc}
>         0 & 0 & 0 & 0\\\\
>         0 & \frac{1}{2} & \frac{\text{i}}{2} & 0\\\\
>         0 & \frac{-\text{i}}{2} & \frac{1}{2}& 0\\\\
>         0 & 0 & 0 & 0
>     \end{array}\right).
> $$
> In this paper, we integrate parity checks and the HWP ansatz to enforce additional constraints and mitigate hardware errors. **We focus on the practical usage of parity checks in near-term quantum computing**, which involves utilizing some good qualities of the HWP ansatz. We have the following distinguishing contributions:
> 1. We proposed a simpler gate, namely the NBS gate, with compatible expressivity as the BS gate.
> 2. We developed a viable method to incorporate parity checks for error mitigation in the HWP ansatz, conducting experiments on both superconducting quantum processors and simulators.
> 3. We proposed a novel approach to use parity checks as projective measurements, thereby restricting the evolving subspace of quantum states. We demonstrated the effectiveness of our method on an NP-hard combinatorial optimization problem. This paradigm holds significant potential for constrained VQE, offering a straightforward means to incorporate hard constraints using simple parity checks.
>
> Therefore, we disagree that "the main contribution is incremental to previous results". According to your review, we think that our work is something more than "For instance, a paper with technical flaws, weak evaluation, inadequate reproducibility and incompletely addressed ethical considerations." We also strongly believe that good incremental work (if you insist) is also within the scope of NeurIPS.
>
> [1] Rethinking the symmetry-preserving circuits for constrained variational quantum algorithms
>
> We will then respond to your questions:
> > Q1: as definition of NBS is kind of constant gate, how to train the HWP anstaz constructed by such fixed gates shown in Fig 1. ? and what is the R in Figure 1 ?
>
> Ans: Thanks for your kind reminder, and we apologize for the misunderstanding caused by our negligence. The matrix provided in the paper is the Hermitian matrix of the NBS gate. The unitary operator of the NBS gate should take exponential of the Hermitian matrix:
> $$U_{NBS}=e^{\text{i}\theta H_{NBS}}=\left(\begin{array}{cccc}
>         1 & 0 & 0 & 0\\\\
>         0 & \frac{\cos\theta+\text{i}\sin\theta+1}{2} & \text{i}\frac{\cos\theta+\text{i}\sin\theta-1}{2} & 0\\\\
>         0 & -\text{i}\frac{\cos\theta+\text{i}\sin\theta-1}{2} & \frac{\cos\theta+\text{i}\sin\theta+1}{2} & 0\\\\
>         0 & 0 & 0 & 1
>     \end{array}\right)=\left(\begin{array}{cccc}
>         1 & 0 & 0 & 0\\\\
>         0 & \frac{e^{\text{i}\theta}+1}{2} & \text{i}\frac{e^{\text{i}\theta}-1}{2} & 0\\\\
>         0 & -\text{i}\frac{e^{\text{i}\theta}-1}{2} & \frac{e^{\text{i}\theta}+1}{2} & 0\\\\
>         0 & 0 & 0 & 1
>     \end{array}\right),$$
>
> which contains trainable parameters. The rotations at the end of the ansatz in Fig.1 are for measuring different decomposed Pauli strings of the problem Hamiltonian. They can be omitted from the ansatz for simplicity.
>
> > Q2: comparing to the proposed HWP, the UCCS not only requires shallow circuit but also have out-performance. What is the advantages of the proposed over it?
>
> Ans: The UCCS ansatz is indeed very shallow, which leads to the "out-performance" on NISQ devices. However, it exhibits relatively weak performance compared to other methods, failing to achieve results within chemical accuracy ($1.6\times 10^{-3}$Ha) for any of the molecules tested. It is essential to consider performance not only during the NISQ era but also beyond. Additionally, other methods can also achieve superior results if similar numbers of layers or parameters are utilized.
>
> > Q3: it may be poor scalable of such ansatz used for combinatorial optimization problem, especially in NISQ era. and in line 252, the probability of finding the feasible state will exponentially small with the problem size, how to extract such feasible state without performing exponentially measurement?
>
> Ans: This one is actually a very good question and we do acknowledge that this might be a potential limitation of the proposed method (as stated in the paper). However, we would like to point out that for an $n$-qubit system, the number of shots we need to obtain the final distribution is $\mathcal{O}(2^n)$, which is why the order of the expectation value remains acceptable. Despite this, the proposed method potentially stands as the best constrained-VQE currently available.

---

> > ### Comment · Reviewer_K5Kf · 2024-08-13
> >
> > Thank you for the response, most of my question are solved. I would happy to raise my score.

---

> > > ### Author Response · Authors · 2024-08-13
> > >
> > > We are sincerely grateful for your careful reading and thoughtful comments on our manuscript, which have been invaluable in enhancing the clarity and rigor of our work. Thank you again for your time and effort in reviewing our paper.
> > >
> > > Best regards

---

> ### Author Response · Authors · 2024-08-11
>
> Dear reviewer K5Kf,
>
> We really appreciate your valuable comments. In our rebuttal, we have highlighted the contributions and answered the questions accordingly. Since the discussion period is approaching its end, we are looking forward to your feedback.
>
> Best regards,
>
> Authors

---

### Author Response · Authors · 2024-08-12

Dear AC:

Thanks for your efforts and the profound suggestions from the reviewers. Since half of the discussion time has passed, we wonder whether the reviewers' concerns have been resolved by our reply and whether they have further questions.

---

### Decision · Program_Chairs · 2024-09-25

**Decision:**

Accept (poster)

**Comment:**

The submission proposes combining the HW-preserving ansatz with qubit topology-aware parity checks to impose hard constraints directly on quantum circuits, which was difficult for implementing important applications in quantum chemistry and optimization (e.g., the Quadratic Assignment problem).  It includes both numerical simulations and experiments on a real quantum device, which further validates the proposed approach. All reviewers acknowledged the authors’ contributions. However, it is suggested that authors should articulate their contributions, especially compared with a few existing references with similar titles.